# Characterizing Preferential Flow Paths in Texturally Similar Soils under Different Land Uses by Combining Drainage and Dye-Staining Methods

**Youyan Zhang [1], Zhe Cao [2,*], Fang Hou [3] and Jinhua Cheng [3]**

[1]   Institute of Desertification Studies, Chinese Academy of Forestry, Beijing 100091, China; youyan_zhang@126.com

[2]   Bureau of Soil and Water Conservation, Water Resources Department of Gansu Province, Lanzhou 730000, China

[3]   College of Soil and Water Conservation, Beijing Forestry University, Beijing 100083, China; CHNhoufang@163.com (F.H.); jinhua_cheng@126.com (J.C.)

\*   Correspondence: caozhe.gansu@foxmail.com or zhecao@sina.com; Tel.: +86-13-893-651-071

**Abstract:** Preferential flow paths have been widely characterized by many visualization methods. However, the differences in preferential flow paths under various land uses and their relationships to hydraulic properties remain uncertain. The objectives of this study are to (1) characterize preferential flow paths under various land uses (forest and orchard) by combining drainage and dye-staining methods and to (2) build a connection between preferential flow paths and hydraulic-related parameters and extract the proportion of preferential flow paths from the compounding effects of matrix flow and preferential flow. The dye-staining experiments were conducted in five sandy soils and one sandy clay loam in situ, including four soils from forest and two soils from orchards. A total of 47 soil cores, 4 cm in height and 9 cm in diameter, were collected in each layer of the dye-stained soils for drainage experiments in the laboratory. Dye coverage and hydraulically equivalent macropore parameters (macroporosity, pore size distribution, and number of macropores) and their relationships were analyzed. The results show that the volume of preferential flow is partly affected by the total macropore volume. The effect of macropores on preferential flow varies by macropore size distribution. Dye coverage exhibited a significant ($P < 0.01$) correlation with macroporosity (correlation coefficient 0.83). Based on the value of macroporosity or steady effluent rates, the part of the dye coverage that was due to preferential flow on the surface dye-stained soil (resulting from both matrix and preferential flow) could be identified in this study. Compared with orchards, forestland has more preferential flow paths in both surface soil and subsoil. Further studies are needed to quantify the 3-D preferential flow paths and build a connection between preferential flow paths and hydraulic properties.

**Keywords:** dye tracer; drainage experiment; macropore size distribution; number of macropores; land use

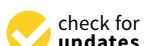



## 1. Introduction

The importance of preferential flow as a transport mechanism of water and solute is widely recognized [1–4]. Preferential flow paths, such as macropores, significantly increase the risk of groundwater contamination [5–9], pesticide contamination, and nutrient loss [10,11]. The preferential paths may promote subsurface flow, which plays an important role in streamflow [6,12] and stormflow generation [10,13], causing soil erosion [14] and even slope failure [15].

Land use and soil type are the main factors that influence the formation and distribution of preferential flow paths. Most studies found that water inputs have lots of effects on preferential flow under different land cover [16], and land use [5]. Cheng [17] reported that vertically oriented preferential flow paths differed significantly under forestland,

shrubland, and farmland. They emphasized that land use has an important effect on the formation of preferential flow paths because root distribution and soil properties are quite different according to the type of land use. Luo [4] found that land use had a significant influence on macroporosity, tortuosity, and hydraulic radius. Within the same soil type, pasture had more developed preferential flow paths compared with row crops, especially in the subsoil. Price [18] noted that soils under natural conditions generally exhibited higher macroporosity compared to soil impacted by human activities.

Dye tracer experiments have been used for preferential path visualization over the past two decades [19] Brilliant blue FCF (C.I. Food Blue 2) remains the tracer of choice in soil water flow experiments due to its mobility, visibility, and toxicity features [7]. As a dye tracer, Brilliant Blue can visually show the distribution of non-uniform flow in soil, that is, preferential flow [20], and dye-stained and unstained areas on dye-stained soil profile photographs can be easily identified for the characterization of preferential flow path distribution [17].

Studies using breakthrough curves were first conducted by Nielsen and Biggar [21] in miscible displacement experiments. Radulovich [22] conducted similar drainage experiments, i.e., the water breakthrough curve, to quantify macropore size distribution and the number of macropores. Macropore sizes obtained from water breakthrough curves are generated from both a basic flow equation and the Poiseuille equation [22]. Water breakthrough curves can be used to calculate pore quantities for different equivalent cylindrical radii. Based on the capillary theory, the final steady effluent rates are controlled by the smallest pore radius in any single continuous flow path [1]. Therefore, the water breakthrough curve could be used to calculate the neck radius [22]. This method may be applied to many different hydraulic-related scenarios.

Although the dye tracer experimental method has advantages for detecting the distribution of preferential flow paths, it suffers from a shortage of quantitative information on hydraulic properties of preferential flow. By considering pore necks, the drainage experimental method is preferred for reflecting real hydraulic properties as the calculated equivalent macroporosity could then be used to build a connection with dye coverage. The primary aim of this paper is to characterize preferential flow paths in texturally similar soils in orchards and forestlands using the two methods. This study also attempts to build a connection between preferential flow paths and hydraulic-related parameters and to extract the proportion of preferential flow paths from the compounding dye-staining coverage that result from matrix flow and preferential flow in surface soil.

## 2. Materials and Methods

### 2.1. Testing Site

The six field sites (29°04′ N, 106°16′ E, two citrus sites; 28°37′ N, 106°24′ E, four forest sites) are located in the central area of Simian Mountain, which is situated in the high-elevation region of the Three Gorges Reservoir Area in southwest China. Southwest China, in particular the upstream area of the Three Gorges Dam, has severe challenges with groundwater security because of the pollution caused by intensive agricultural activities. Such activities have included dispersing large amounts of fertilizers and pesticides during periods of vegetative growth to sustain and increase food and timber yields.

The research sites ranged in elevation from 300 m to 1500 m above sea level. Simian Mountain is located in a sub-tropical area that is characterized by a continental monsoon climate and abundant precipitation. The average annual temperature is 13.7 °C, with the lowest monthly average temperature reaching −5.5 °C in January and the highest temperature reaching 31.5 °C in August. The average annual precipitation is 1522.3 mm. Temporal variation of precipitation follows seasonal patterns and the highest levels occur during the summer months. The field study was conducted from May to September of 2012. Six field sites were investigated, including two hardwood forest sites, two mixed forest sites (hardwood and softwood), and two citrus orchard sites (Table 1).

**Table 1.** Basic sites information.

| Types [1] | Sites Code | Elevation (m) | Slope Aspect | Soil TEXTURE |
|:---:|:---:|:---:|:---:|:---:|
| CO | A | 300 | north | sandy loam (clay loam) [2] |
| | B | 304 | north | sandy loam |
| HF | C | 1182 | north | sandy loam |
| | D | 1183 | northwest | sandy loam |
| MF | E | 1172 | northwest | sandy loam |
| | F | 1173 | northeast | sandy loam |

[1] CO, citrus orchard sites; HF, hardwood forest sites; MF, mixed forest. [2] The soil texture in surface soil (10 cm) of site A is clay loam.

### 2.2. Dye—Staining Experiment and Image Processing

Dye tracer experiments were conducted at the six field sites using the methods described by Cheng [17]. Prior to the experiments, fallen leaves covering the experimental sites were removed. A flat area of 100 cm by 100 cm was selected at each site for the dye tracer experiments. A 70 cm × 70 cm square iron frame (height 50 cm, thickness 0.5 cm) was embedded in the middle of the area at a depth of 30 to 40 cm. Surface soil layers within 5 cm of both sides of the iron wall were compacted to prevent dye tracers from infiltrating along the frame wall. Water (10 L) mixed with brilliant blue dye at a concentration of $4 \text{ g L}^{-1}$ was sprayed onto the soil surface. The spray rate depended on the soil infiltration rates in situ to avoid pooling. Twenty-four hours after spraying the soil, a trench was created next to the dye application area to capture a vertical cross-sectional image. The experimental plot was then dug up in consecutive, 10 cm deep horizontal cross-sectional soil layers (SL) (0–10 cm soil layer as 1st SL, 10–20 cm soil layer as 2nd SL, and 20–30 cm soil layer as 3rd SL) (Figure 1).

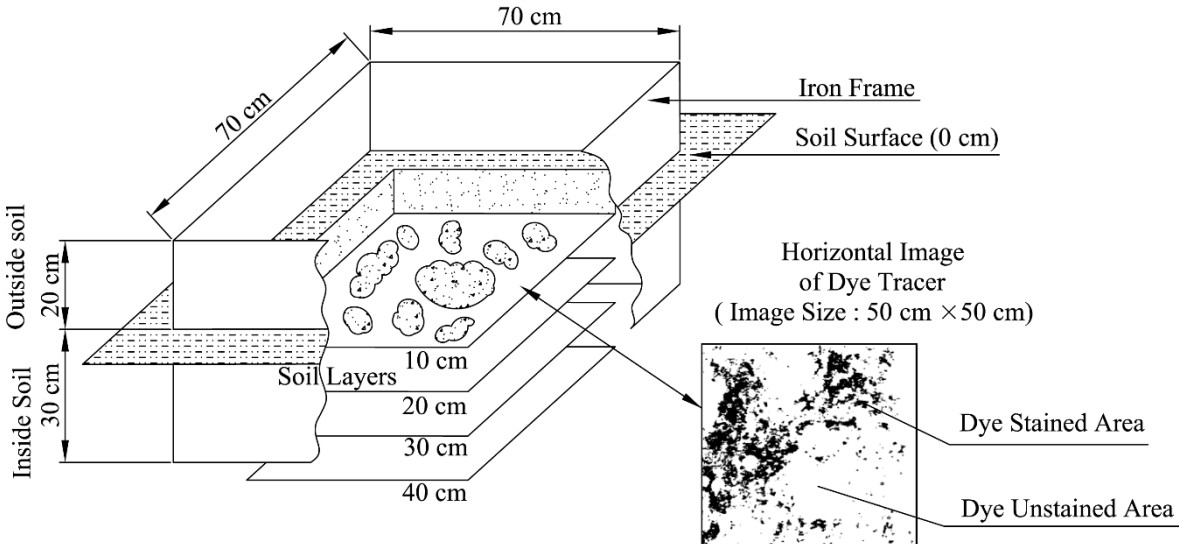

**Figure 1.** Sketch map of the soil profile in brilliant blue dyeing experiment.

The image of each horizontal cross section was captured. Vertical and horizontal cross-sectional images were captured using a digital camera with a resolution of 2560 by 1920 pixels. Image analysis geometrically corrected the captured images by using Photoshop CS3. The sizes of the vertical sections of the pictures were cropped to 50 cm × 50 cm by referring to the image ruler. Then, the saturation, brightness, greyscale, and threshold values were adjusted. The dye-stained sections were then replaced by black and gray while the unstained areas were replaced by white. The color-replaced images were then imported into Image-Pro Plus (version 6.0 Media Cybernetics, Inc., Rockville, MD, USA, 2006) to calculate the RGB values, which black was (255,255,255) and white was (0,0,0). The RGB

values were exported to Microsoft Excel files and used to calculate the dye coverage. A total of 44 images were processed, including 26 images from the horizontal cross section of soil and 18 images from the vertical cross section of soil. We could calculate the preferential flow indices by using Sigmaplot [17] to calculate these binary matrices (Figure 2).

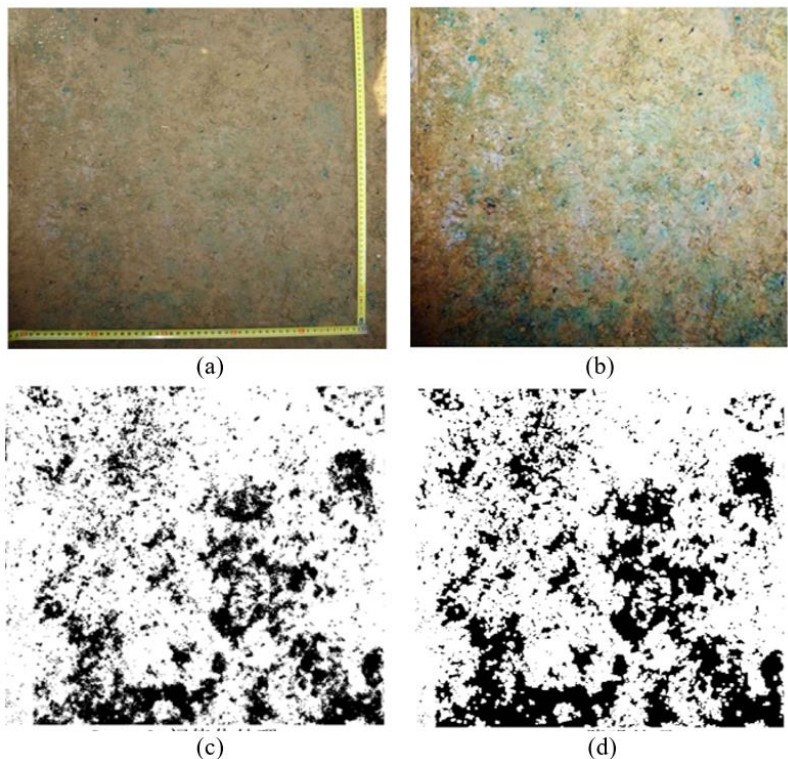

**Figure 2.** Image processing of soil dyeing. The photos were processed in Photoshop CS3 (Adobe Systems Inc., San Jose, CA, USA) and Image Pro Plus 6.0 (Media Cybernetics Inc., Rockville, MD, USA) to quantify the stained areas [17]. (**a**) Step 1: Original image; (**b**) Step 2: Illumination correction; (**c**) Step 3: Brightness, greyscale and threshold values adjusting; (**d**) Step 4: Binary processing.

*2.3. Drainage Experiment*

Iron rings of 4 cm in height and 9 cm in diameter were embedded into the soil to collect complete soil samples from each of the 10 cm-deep layers. For each layer, two core samples were collected, one from the dye-stained area and another from the unstained areas [17]. In total, 47 core samples were collected. Embedded iron ring locations were marked on the horizontal cross-sectional images to obtain dye coverage values for the soil samples. To prepare the soil cores for the drainage experiments, the cores were saturated and free drained for 12 h.

Drainage experiments were conducted in a laboratory for a duration of 200 s. Each soil sample was transferred to a copper ring with a water inlet and water outlet, and the copper ring then was sealed with a cap. The water inlet was connected to a constant pressure bottle by a tube. The constant pressure bottle allowed air to enter as water drained, thereby maintaining a constant water pressure at the water inlet. Macropores were defined as pores that drain when water breakthrough curves reach steady state flow, which corresponds to the matrix potential section ranging between 0 and $-3$ kPa in Radulovich [22]. The theory applied in this drainage experiment is consistent with that of Radulovich [22]. The equivalent radii are calculated from the measured soil outflow rates. The maximum macropore radius was calculated when the effluent water first appeared, while the smallest macropore radius was calculated when the effluent water reached a steady state. By applying the water breakthrough curve, the first effluent cut-off time was recorded during the experiment, and the steady flow time was determined visually (Figure 3). Radii were calculated [22] by combining a basic flow equation

$$Q = AJ_w = \frac{\pi r^2 Lc}{t} \tag{1}$$

and the Poiseuille equation

$$Q = \frac{\pi r^4 \Delta p}{8\eta L_c} \tag{2}$$

with the pore radius

$$r = L_C \sqrt{\frac{8\eta}{t\Delta P}} \tag{3}$$

where $Q$ is the flow discharge (cm$^3$ s$^{-1}$), $A$ is the pore section area (cm$^2$), $Jw$ is the flow velocity in the flow channel (cm s$^{-1}$), $Lc$ is the core lengths ($L$) multiplied by the tortuosity ($\tau$) (here assigned a value of 1.2), t is the time when water was initially added (s), $\Delta p$ is the pressure head (cm), and $\eta$ is viscosity (g cm$^{-1}$ s$^{-2}$) [22].

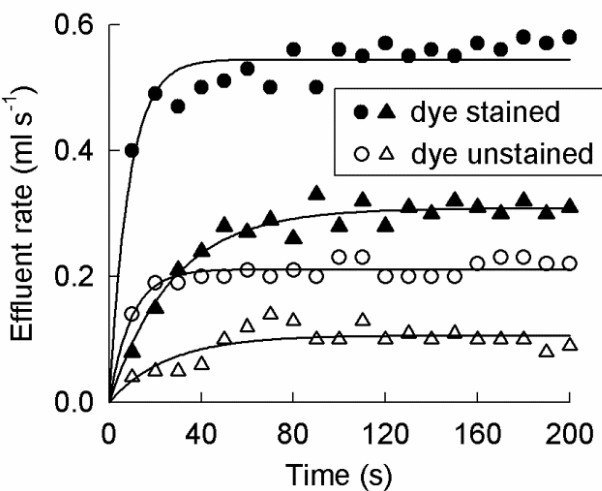

**Figure 3.** Typical curves of water breakthrough. Time on the abscissa is the duration of the water breakthrough. The time of the first effluent determines the largest macropore radius using Equation (3). All the curves in this figure are from the orchard (A site). The black circle indicates the 1st soil layer, the black triangle indicates the 2nd soil layer, the open circle indicates the 1st soil layer, and the open triangle indicates the 2nd soil layer.

The pore radii could be calculated from the effluent time. The volume of effluent water was measured at 5 s intervals. The range of pore radii was divided arbitrarily into 0.2–0.4-mm intervals, and the median radii were then calculated. The increments between adjacent interval effluent volumes ($Qe$) were calculated. Finally, the number of average radius pores ($n$) could be calculated by

$$Q = nAJ_w \tag{4}$$

### 2.4. Statistical Analysis

The bivariate correlations (the correlation coefficient was the default Pearson coefficient) and linear regression analyses were carried out using the IBM SPSS Statistics software package (version 19.0, IBM Corp., Chicago, IL, USA, 2010).

### 3. Results and Discussion

#### 3.1. Preferential Flow Paths and Dye-Staining Coverage

Preferential flow paths were visualized by dye, as shown in Figures 4 and 5. Dye-stained proportions of the 1st soil layers in the A and B sites were greater than 80% (Figure 4). This result is similar to the results obtained by Öhrström [23], who reported dye coverage levels of close to 100% in the upper 5 to 10 cm soil layer. The dye coverage

in the 2nd soil layer in the A and B sites dropped to 7.25 and 15.39%, respectively. Dye coverage decreased rapidly with depth, which was also found in the studies by Cheng [17] and Öhrström [23]. Matrix flow contributed to more dye-stained soil in the surface soil than in the lower soil layers. In site B, only 0.32% of the soil area in the 3rd soil layer was stained, as shown in Figure 5f This result indicates that few preferential flow paths travelled vertically through this layer. The horizontally oriented preferential flow paths in the four forest sites (Figure 5c–f) mainly reached a depth of 45 cm, which was almost in accordance with the depth of 40 cm reported by Flury and Flühler [19]. Distribution depths of the preferential flow paths most likely differed because of the differences in soil type and land uses [24]. Soil that is more structured and less disturbed always has more developed preferential flow paths. In this study, the subsoil of the forest land clearly had more developed preferential flow paths than the orchard land.

Dye-staining patterns were different between soil layers. At site A, few horizontally oriented paths of preferential flow were found in the 2nd and 3rd soil layers, and the lower layers of the vertical profiles were nearly no dyeing paths (Figure 5a). When considering the total number of preferential flow paths for a certain site, fewer horizontally oriented paths of preferential flow indicate, to some extent, greater vertically oriented preferential flow paths. Consequently, well-developed vertical paths of preferential flow were found in site A. The preferential flow paths exhibited a concentrated pattern of horizontal distribution in the 2nd soil layer (Figure 4c) and a punctiform distribution in the 3rd soil layer (Figure 4d). There was significantly more stained soil in the 2nd soil layer than in the 3rd soil layer. For site B, 0.32% of the soil area in the 3rd soil layer was stained, and the paths of preferential flow were well-developed along the horizontal orientation (Figure 5b). This outcome may have been caused by lateral roots running beneath the surface soil [25,26] or animal activities in this layer [5,25,27]. In general, the distribution of preferential flow paths appeared to be random.

The front line of dye in the vertical cross-sectional images (Figure 5) fluctuated to depths of 3 cm and at least 9 cm for orchards and forests, respectively. Dye-stained areas near the soil surface were partly attributed to the effect of matrix flow. The depths of the front line fluctuation mainly reflect the influence of the edge of the matrix flow. This result indicates that the forest soils had better water infiltration than orchard soils. Based on the drainage order, the presence of macropores in the upper layers may dominate water conductivity processes. Previous studies have shown that macropores only marginally contribute to total soil porosity [1,22]. In our study, dye coverage was indeterminate below a specific soil depth because the dye did not penetrate that far. This result is similar to the results of Flury [24], who found this depth to be at least 10 cm. In addition, preferential flow can be initiated from both topsoil and a saturated part of the subsoil [28], thus affecting the movement of preferential flow. The proportion of preferential flow paths must be extracted from the compounding dye-staining results, which include both preferential flow and matrix flow.

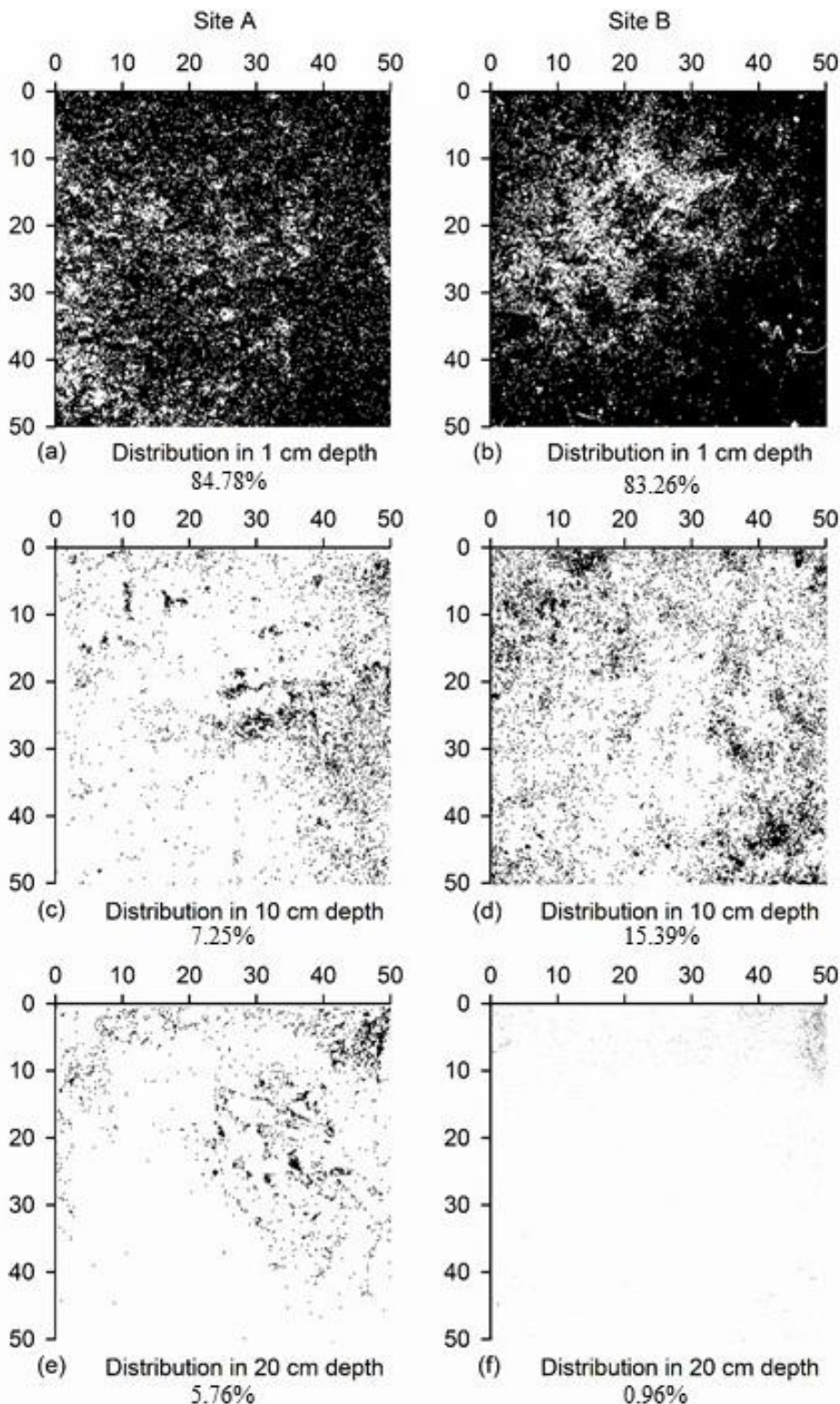

**Figure 4.** Horizontal dye-stained soil profile in citrus sites (cm). (**a–f**) correspond to the horizontal profile of citrus site in 1 cm, 10 cm and 20 cm depth, respectively. The percentages below represent the dyed area ratio of each profile.

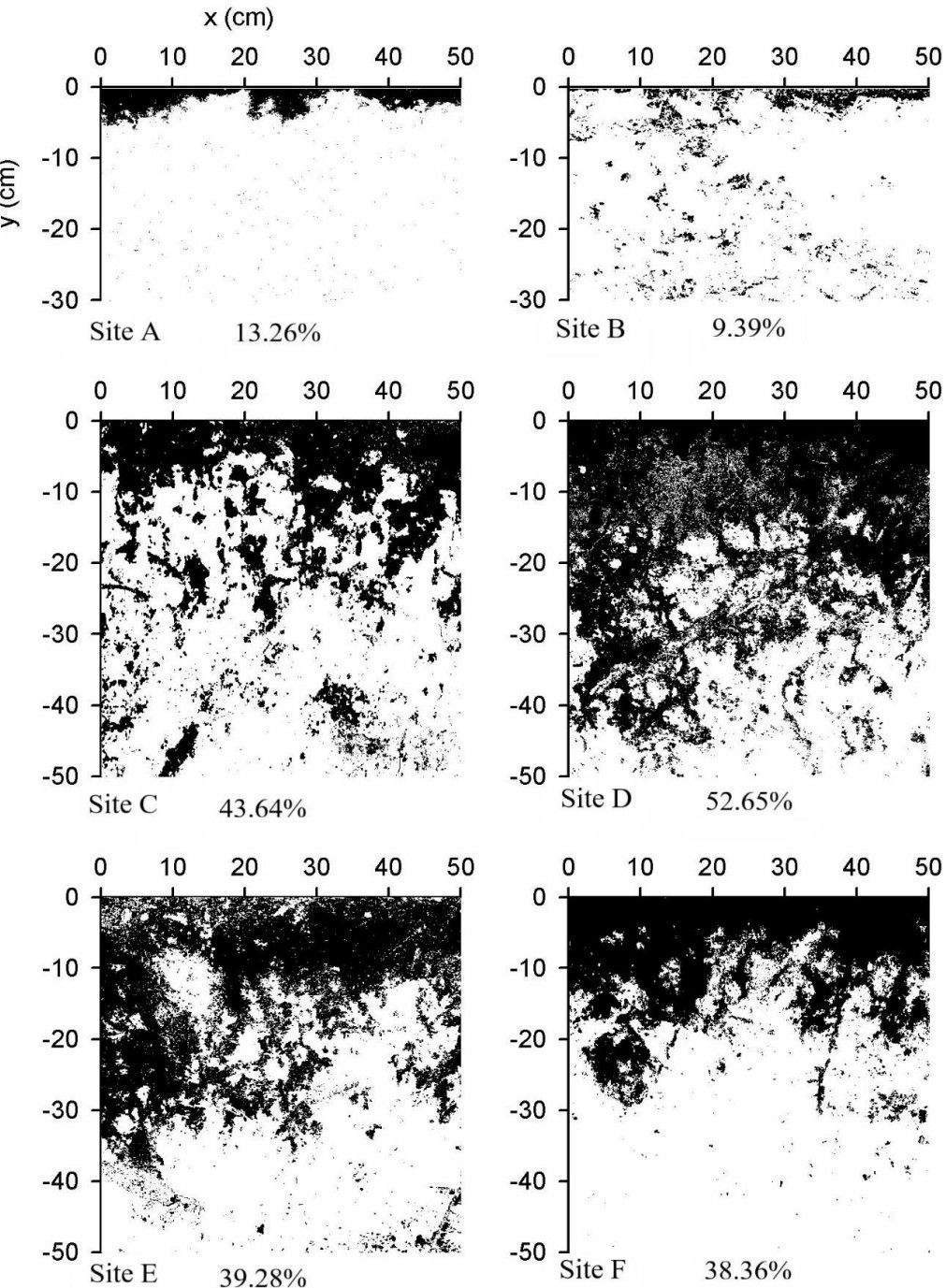

**Figure 5.** Vertical dye-stained soil profile (cm). Site A and Site B, Site C and Site D, Site E and Site F correspond to the vertical profile of citrus orchard sites, hardwood forest sites and mixed forest, respectively. The percentages below represent the dyed area ratio of each profile.

*3.2. Water Breakthrough Curves*

The steady effluent rates and ratios are shown in Table 2. All of the steady effluent rates in the dye-stained sections were clearly higher than those in the unstained sections. This result provides direct evidence that preferential flow paths (macropores) increased the effluent rates and may have produced high levels of water conductivity in the soil sites. The ratios reflect the contribution that effective continuity macropores make to the steady effluent rate. Higher ratios indicate cases in which macropores were significant and

contributed to the water conductivity values, especially when such ratios were found for deeper soil.

**Table 2.** Steady effluent rate and ratio.

| Soil Depth (cm) | Dye Section | Steady Effluent Rate (mL s$^{-1}$) | | | | | | Effluent Ratio of ds to dus (%) | | | | | |
|---|---|---|---|---|---|---|---|---|---|---|---|---|---|
| | | CO | | HF | | MF | | CO | | HF | | MF | |
| | | A | B | C | D | E | F | A | B | C | D | E | F |
| 0–10 | ds | 0.56 | 0.34 | 0.31 | 0.48 | 1.02 | 0.43 | 1.90 | 1.54 | 2.36 | 2.00 | 2.41 | 0.54 |
| | dus | 0.29 | 0.22 | 0.13 | 0.24 | 0.42 | 0.79 | | | | | | |
| 10–20 | ds | 0.48 | 0.55 | 0.88 | 0.26 | 0.26 | 0.39 | 2.23 | 2.20 | 4.22 | 1.96 | 2.28 | 12.11 |
| | dus | 0.21 | 0.25 | 0.21 | 0.13 | 0.11 | 0.03 | | | | | | |
| 20–30 | ds | 0.30 | | 0.73 | 0.11 | 0.55 | 0.43 | 2.82 | | 3.08 | 0.89 | 2.01 | 8.67 |
| | dus | 0.11 | 0.10 | 0.24 | 0.13 | 0.27 | 0.05 | | | | | | |
| 30–40 | ds | / | / | 0.15 | 0.20 | 0.26 | 0.23 | | | 1.17 | 1.06 | 0.50 | 7.41 |
| | dus | / | / | 0.13 | 0.19 | 0.52 | 0.03 | | | | | | |
| 40–50 | dus | / | / | 0.04 | 0.26 | 0.22 | 0.11 | | | | | | |

ds, dye stained section; dus, dye unstained section.

The breakthrough curves exhibited a stronger fluctuation in the dye-stained section than in the unstained section (Figure 3). This stronger fluctuation in breakthrough curves was consistent with the results of Radulovich [22]. Compared to the unstained section, the effluent rate stabilized in the dye-stained section at a later time, partly due to its susceptibility to pressure potential. Capillary theory may provide an explanation for this phenomenon. Water flux generated by saturated capillaries is orders of magnitude larger than that in unsaturated capillaries. Similarly, water flux in saturated soils is more sensitive to potential energy [27,29,30] and macropore connectivity (continuity of capillaries) [15,27,30] than that in unsaturated soil. Additionally, the flow inertia and viscosity force may destroy preferential flow paths; however, wetter conditions may facilitate interconnection between macropores [10,15,16], causing a change in connectivity and thereby affecting the effluent rate and steady time of breakthrough curves. In other words, minor changes in soil aeration can result in a major change in water flux, especially for the preferential flow paths. This effect accounted for the stronger fluctuation of breakthrough curves in the dye-stained section.

*3.3. Effectiveness of Hydraulically Equivalent Macropore*

The quantitative macropore size and number were generated from the measured effluent volume. Hydraulic equivalent macropore radii ranged between 0.3 and 1.7 mm in the CO sites (Table 3) and 0.3 to 2.7 mm in the HF and MF sites. For most radius divisions, the number of macropores in the dye-stained sections is much larger (in some cases, at least an order of magnitude) than in the unstained sections because the macropores are channels. The number of macropores and the area of dye coverage showed the same declining trend as with soil depth. Importantly, there were macropores in both the dye-stained and unstained sections (Table 3), although the pore radii in the dye-stained sections were larger than those in the unstained sections. This result may be partly due to the connectivity of the preferential flow paths and suggests that macropores with larger pore radii may have greater priority of preferential flow, which can lessen the (potential) water conductivity of macropores in unstained sections. This suggestion is also confirmed by the fact that within the same radius range, there was almost the same magnitude of macropores in the unstained sections as in the dye-stained sections. However, in the largest radius range, the dye-stained section had more macropores than the unstained sections. For example, for pore radii of 0.3–0.7 mm (Table 3), the unstained section of the 3rd soil layer in the B site (2.01 × 104) had almost an identical number of macropores as the dye-stained section of the 3rd soil layer in the A site (2.00 × 104). As another example, the unstained section had a larger effluent rate than the dye-stained section of the 3rd soil layer in the D site. In

this case, more macropores in the largest radius range led to the dye staining. Generally, macropores with larger radii may account for more preferential flow.

**Table 3.** Composition of macropore size and number in citrus sites.

| Sites | Soil Layers | Sections | Number of Macropore in Size Range (m$^{-2}$) | | | | | Macroporosity (%) |
|---|---|---|---|---|---|---|---|---|
| | | | 1.4–1.7 mm | 1.2–1.4 mm | 1.0–1.2 mm | 0.7–1.0 mm | 0.3–0.7 mm | |
| A | 1 | ds | 0 | 23 | $1.80 \times 10^2$ | $1.52 \times 10^3$ | $5.16 \times 10^4$ | 4.48 |
| | | dus | 0 | 0 | $0.05 \times 10^2$ | $1.20 \times 10^3$ | $2.21 \times 10^4$ | 2.01 |
| | 2 | ds | 1 | 14 | $0.63 \times 10^2$ | $2.06 \times 10^3$ | $7.59 \times 10^4$ | 6.47 |
| | | dus | 0 | 0 | 0 | $0.36 \times 10^3$ | $2.12 \times 10^4$ | 1.74 |
| | 3 | ds | 0 | 0 | 0 | $0.37 \times 10^3$ | $2.01 \times 10^4$ | 1.66 |
| | | dus | 0 | 0 | 0 | 0 | $1.19 \times 10^4$ | 0.94 |
| B | 1 | ds | 0 | 0 | $0.56 \times 10^2$ | $1.35 \times 10^3$ | $1.49 \times 10^4$ | 1.50 |
| | | dus | 0 | 0 | 0 | $0.58 \times 10^3$ | $0.54 \times 10^4$ | 0.55 |
| | 2 | ds | 7 | 28 | $1.07 \times 10^2$ | $2.36 \times 10^3$ | $7.65 \times 10^4$ | 6.60 |
| | | dus | 0 | 0 | 0 | $0.70 \times 10^3$ | $0.80 \times 10^4$ | 0.79 |
| | 3 | dus | 0 | 0 | 0 | 0 | $2.00 \times 10^4$ | 1.57 |

The different effluent rates may be linked to the differences in potential energy [19,27,30], continuity [10,15,26], and tortuosity [26,31,32], and this relationship may be reflected by the various pore size distributions [29]. The preferential effects of macropores on water conductivity are not constant across macropore size distributions, even if the pore radii were the same or in the same range. Even in the unstained sections of forest soils, there were more macropores than in orchard soils, indicating that the effectiveness of macropores may vary with macropore size distributions, and confirms that macropores with larger radii may account for more preferential flow. In other words, a correlation was found between the steady effluent rate and macropore characteristics. Further bivariate correlation analysis showed that both macroporosity and macroporosity for pore radii > 0.7 mm were significantly ($P < 0.01$) related to the steady effluent rate, and Pearson correlation values were found to be 0.94 and 0.96, respectively. A linear regression (Figure 6) shows that macroporosity for pore radii > 0.7 mm accounted for 92% of the variation in the steady effluent rate, in contrast to 89% of the variation explained by macroporosity.

*3.4. Relationship between Dye Coverage and Hydraulically Equivalent Macroporosity*

Data from dye-stained profiles were used to establish the relationship between dye coverage and hydraulically equivalent macroporosity, despite the images of the 1st soil layers showing that the dye-stained areas were notably influenced by matrix flow. A Pearson correlation coefficient of 0.83 identified that macroporosity was significantly ($P < 0.01$) related to dye coverage. Linear regressions (Figure 7) showed that macroporosity may explain 69% of the dye coverage variation, and macroporosity with pore radii > 0.7 mm can explain 81% of dye coverage variation. These results show that there is a relationship between macroporosity (hydraulically equivalent) and dye coverage (visible preferential paths). Macroporosity for pore radii > 0.7 mm accounted for more variations in effluent rates and dye coverage. This finding suggests that larger macropores have a significant effect on the hydraulic properties and path distributions of preferential flow.

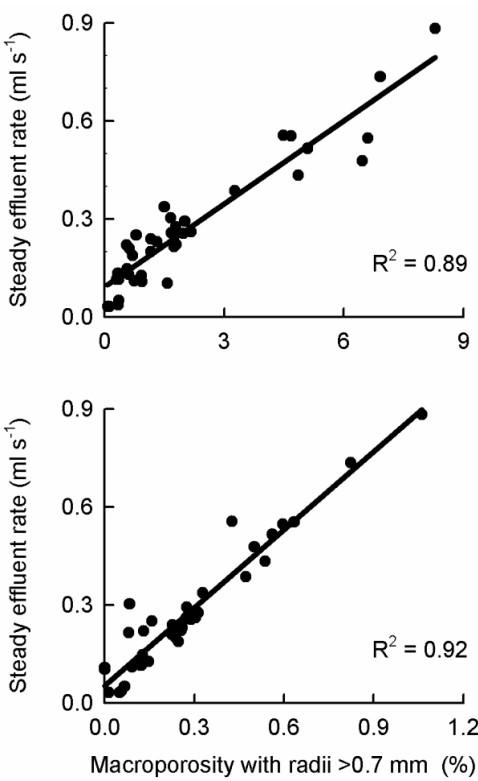

**Figure 6.** Relationship between steady effluent rate and macroporosity, and macroporosity (>0.7 mm).

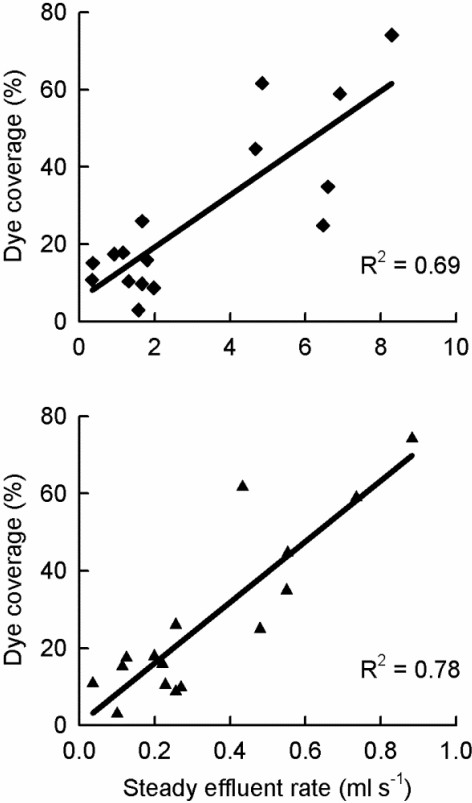

**Figure 7.** Relationship between dye coverage and macroporosity, and steady effluent rate.

The dye-stained coverage in surface soil was attributed to both the matrix flow and the preferential flow. Based on the linear relationship between dye coverage and macroporosity

for pore radii > 0.7 mm, the preferential flow contributed 26.5%, 28.3%, and 65.7% dye coverage in the 1st soil layers for orchards, hardwood, and mixed wood, respectively. Consequently, the dye coverage is due to preferential flow in surface soil.

The laboratory results also showed a limitation in scale extrapolation. In particular, the sampling cylinder used in the water breakthrough experiments had a small volume. The small size inevitably caused some horizontally or vertically oriented preferential flow paths to be cut off, which damaged the lateral connectivity and changed the tortuosity and continuity. Therefore, larger sample volumes are needed for the drainage experiments to match the scale of the dye-staining results. Further studies are needed to build a connection between 3-D preferential flow paths and hydraulic properties.

## 4. Summary and Conclusions

The matrix flow may reach a depth of 3 cm for orchard soils and at least a depth of 9 cm for forest soils. Compared to orchards, forestlands exhibited more developed preferential flow paths in both surface soil and subsoil. The effect of macropores on preferential flow changes with different macropore size distributions. Macropores with larger radii have greater impact on preferential flow than macropores with smaller radii. Dye coverage exhibited a significant relationship with macroporosity (correlation coefficient of 0.83). Based on the value of macroporosity or steady effluent rates, the part of the dye coverage that was due to preferential flow on the surface dye-stained soil (resulting from both matrix and preferential flow) could be identified in this study.

**Author Contributions:** Conceptualization, Y.Z., J.C., Z.C., and F.H.; methodology, Y.Z.; software; validation, Y.Z., J.C., and F.H.; formal analysis, Y.Z.; investigation, Y.Z., J.C., and F.H.; resources, Y.Z.; data curation, Y.Z.; writing—original draft preparation, Y.Z.; writing—review and editing, Y.Z.; visualization, Y.Z.; supervision, Y.Z.; project administration, J.C.; funding acquisition, J.C. All authors have read and agreed to the published version of the manuscript.

**Funding:** This research was funded by Key techniques for rapid vegetation restoration on degraded forest land, grant No. 2019YFF0303203-3 and National Natural Science Foundation of China, grant No. 32071839.

**Institutional Review Board Statement:** Not applicable.

**Informed Consent Statement:** Not applicable.

**Data Availability Statement:** Not applicable.

**Acknowledgments:** Financial assistance for this work was provided by Key techniques for rapid vegetation restoration on degraded forest land, grant No. 2019YFF0303203-3 and the General Program of the National Natural Science Foundation of China under grant No. 41271300. Special thanks to John L. Nieber and the article reviewers for their helpful comments.

**Conflicts of Interest:** The authors declare no conflict of interest.

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
