# Peer review of "Characterizing Preferential Flow Paths in Texturally Similar Soils under Different Land Uses by Combining Drainage and Dye-Staining Methods"

_water, doi:10.3390/w13020219_

Round 1

Reviewer 1 Report

The authors have addressed most of my previous comments. Hence I can recommend the publication of this manuscript in its current form.

Author Response

Thanks again for your comments on our manuscript.

Reviewer 2 Report

I really like the paper as I said before. Thanks for including the diagram of your experimental set up. It makes everything clearer.

The abstract still contains a statement saying that the six soils are similar. I said in my original review: “The six soils are not similar. There are 5 sandy loams and one clay loam. That would make them behave very differently. I still think that you should change the abstract to say that you had 5 sandy soils and one sandy clay loam.

Please check on the spelling of Poiseuille. I think you forgot the second ‘I’. It’s a French name and the combination of letters “euille” gives it a different sound which was constant trouble for me with my French teacher many years ago.

Line 223: The meaning of the word “indeterminate” is still not clear. Does that mean that it could not be determined because the dye did not penetrate that deep? You may want to clarify this in the paper with something like “,i.e. the dye did not penetrate that far”. Or with a sentence that makes it clear what you mean by indeterminate.

Line 223: when you say “above a specific oil depth”, do you mean deeper in the soil (i.e. greater value of depth) or do you mean closer to the surface? If it is deeper, then you need to use the word “below” rather than ‘above”.

Figure 3 caption: ‘blank’ should be replaced by ‘open’ for both the triangle and circle.

One question I had was by how much your results deviated from Poiseuille’s law?

Author Response

Point 1: I really like the paper as I said before. Thanks for including the diagram of your experimental set up. It makes everything clearer.

Response 1: Thank you for your positive comments, and we have revised the manuscript point by point carefully.

Point 2: The abstract still contains a statement saying that the six soils are similar. I said in my original review: “The six soils are not similar. There are 5 sandy loams and one clay loam. That would make them behave very differently. I still think that you should change the abstract to say that you had 5 sandy soils and one sandy clay loam.

Response 2: Thanks for your suggestion. The sentence has been revised as:    “The dye-staining experiments were conducted in five sandy soils and one sandy clay loam six soils with similar texture in situ, including four soils from forest and two soils from orchards.” (Line 19-21)

Point 3: Please check on the spelling of Poiseuille. I think you forgot the second ‘I’. It’s a French name and the combination of letters “euille” gives it a different sound which was constant trouble for me with my French teacher many years ago.

Response 3: Thanks for your suggestion. “Poiseulle” has been revised as “Poiseuille”. (Line 65 and Line 159)

Point 4: Line 223: The meaning of the word “indeterminate” is still not clear. Does that mean that it could not be determined because the dye did not penetrate that deep? You may want to clarify this in the paper with something like “,i.e. the dye did not penetrate that far”. Or with a sentence that makes it clear what you mean by indeterminate.

Response 4: Thanks for your comment. The “indeterminate” means that it could not be determined because the dye did not penetrate that deep sometimes. In order to clarify this in paper, we have been revised as “dye coverage was indeterminate above below a specific soil depth because the dye did not penetrate that far.” (Line 223-224)

Point 5: Line 223: when you say “above a specific oil depth”, do you mean deeper in the soil (i.e. greater value of depth) or do you mean closer to the surface? If it is deeper, then you need to use the word “below” rather than ‘above”.

Response 5: Thanks for your suggestion. What we say “above a specific oil depth” means deeper in the soil, and “above” has been revised as “below”. (Line 223- 224)

Point 6: Figure 3 caption: ‘blank’ should be replaced by ‘open’ for both the triangle and circle.

 Response 6: Thanks for your suggestion. “blank” has been revised as “open”. (Line 178- Line 179)

Point 7: One question I had was by how much your results deviated from Poiseuille’s law?

Response 7: Macropore sizes obtained from water breakthrough curves are generated from both a basic flow equation and the Poiseuille equation. The dye coverage rate was obtained by dye tracer experimental method. The primary aim of this paper is to characterize preferential flow paths using the two methods. Just as the manuscript “Dye coverage exhibited a significant relationship with macroporosity (correlation co-efficient of 0.83). ”(Line 334-335)
